# Automatic Detection of Acute Leukemia (ALL and AML) Utilizing Customized Deep Graph Convolutional Neural Networks

**DOI:** 10.3390/bioengineering11070644

**Published:** 2024-06-24

**Authors:** Lida Zare, Mahsan Rahmani, Nastaran Khaleghi, Sobhan Sheykhivand, Sebelan Danishvar

**Affiliations:** 1Biomedical Engineering Department, Faculty of Electrical and Computer Engineering, University of Tabriz, Tabriz 51666-16471, Iran; lidazare@tabrizu.ac.ir (L.Z.); mahsanrahmani@tabrizu.ac.ir (M.R.); nastarankhaleghi@tabrizu.ac.ir (N.K.); 2Department of Biomedical Engineering, University of Bonab, Bonab 55517-61167, Iran; s.sheykhivand@tabrizu.ac.ir; 3College of Engineering, Design and Physical Sciences, Brunel University London, Uxbridge UB8 3PH, UK

**Keywords:** ALL, AML, deep learning networks, leukemia, graph

## Abstract

Leukemia is a malignant disease that impacts explicitly the blood cells, leading to life-threatening infections and premature mortality. State-of-the-art machine-enabled technologies and sophisticated deep learning algorithms can assist clinicians in early-stage disease diagnosis. This study introduces an advanced end-to-end approach for the automated diagnosis of acute leukemia classes acute lymphocytic leukemia (ALL) and acute myeloid leukemia (AML). This study gathered a complete database of 44 patients, comprising 670 ALL and AML images. The proposed deep model’s architecture consisted of a fusion of graph theory and convolutional neural network (CNN), with six graph Conv layers and a Softmax layer. The proposed deep model achieved a classification accuracy of 99% and a kappa coefficient of 0.85 for ALL and AML classes. The suggested model was assessed in noisy conditions and demonstrated strong resilience. Specifically, the model’s accuracy remained above 90%, even at a signal-to-noise ratio (SNR) of 0 dB. The proposed approach was evaluated against contemporary methodologies and research, demonstrating encouraging outcomes. According to this, the suggested deep model can serve as a tool for clinicians to identify specific forms of acute leukemia.

## 1. Introduction

Leukemia is a hematologic malignancy originating in the bone marrow, characterized by the excessive generation of abnormal blood cells [1,2]. Leukemia presents flu-like symptoms such as bleeding, bruising, bone pain, and fever. Leukemia can lead to infection within the body and, in certain instances, result in untimely mortality [3]. Generally, this disease is characterized by an elevated count of aberrant blood cells relative to normal cells, leading to the uncontrolled growth of leukocytes [4]. This condition can be diagnosed at any age but is more commonly detected in those under 15 and over 55 [5].

Based on 2018 reports, the United States alone has seen over 60,000 new cases of leukemia, which makes up about 3.5% of all cancer cases in the country. Leukemia is categorized into four distinct types: AML, ALL, chronic myeloid leukemia (CML), and chronic lymphocytic leukemia (CLL), as depicted in Figure 1. This disease can spread to other organs, including the spleen, brain, liver, and kidney, by traveling through the circulation [6,7,8]. Leukemia is diagnosed through a blood test or a biopsy involving bone marrow sampling. Following the blood test, the pertinent pathologist examines the blood-sample under the microscope and assesses the blood samples by analyzing their morphology [9,10]. Therefore, the determination of a leukemia diagnosis relies on the pathologist’s expertise, experience, and level of weariness [11]. Visual diagnosis has low accuracy due to the resemblance between healthy and diseased blood samples, requiring a significant amount of time to complete. The rapid detection of this disease is crucial to prevent the deterioration of the patient’s health. Hence, an automated approach to diagnosing leukemia was developed [11]. In light of this, recent research has focused on developing automated methods for leukemia detection, which will be examined in the following sections.

Zhou et al. [12] introduced a comprehensive leukemia diagnosis system that relies on deep learning networks. The researchers employed CNNs to carry out feature selection/extraction and classification. The solution devised by these researchers utilized the end-to-end approach and did not necessitate any further pre-processing. Their proposed approach achieved an ultimate accuracy of approximately 85%. One of the drawbacks of this research was the limited precision in classification. Khandkar et al. [13] employed the ALL-IDB1 and CNMC 2019 databases to categorize two distinct types of leukemia using machine learning networks. The researchers employed the thresholding approach to classify data and attained an impressive accuracy rate of 95%. One of the drawbacks of this research was the high level of computational complexity associated with the approach. Chola et al. [14] used deep learning to identify and classify eight types of blood cells: basophils, eosinophils, erythroblasts, immature granulocytes, lymphocytes, monocytes, neutrophils, and platelets. The researchers compared their suggested model against pre-learned networks such as DenseNet, ResNet, Inception, and MobileNet and reached a maximum accuracy of 98%. One of the benefits of this study was the presentation of the eight-class situation, while one of the drawbacks was the sizeable computational volume. Bhute et al. [15] employed deep learning networks to categorize acute leukemia. The researchers utilized pre-trained networks to train their model. The researchers used pre-trained networks such as Inception V3, ResNet50, and VGG16.

The classification achieved a remarkable accuracy of 90%. One advantage of this research was its low computer complexity, whereas one disadvantage was the small database available for evaluation. Rastogi et al. [16] proposed a new two-step approach for classifying leukocytes in the diagnosis of acute leukemia. Their proposed model was built on Leufeutx, which is a modification of pre-trained VGG networks. These researchers acquired a detection accuracy of approximately 96% using the ALL-IDB2 database. Dese et al. [17] proposed a machine learning-based automatic diagnosis system for acute leukemia. Their approach was capable of categorizing four common forms of leukemia. One of the benefits of this study was the ability to achieve 95% classification accuracy, albeit the restricted number of classes in the experiment can be considered a drawback.

Ansari et al. [18] utilized deep learning networks to automate the identification of acute leukemia. Using a combination of deep convolutional networks and type 2 fuzzy functions, the researchers could effectively differentiate between the two categories of AML and ALL. The use of this method resulted in an accuracy rate that surpassed 90%. Binary classification is one of the research problems. Abhishek et al. [19] introduced a novel database consisting of 750 images derived from blood microscopic smears. The researchers’ collection comprised photos of chronic lymphocytic leukemia, acute lymphoblastic leukemia, chronic myeloid leukemia, and acute myeloid leukemia. A modified VGG16 pre-trained network was utilized to classify these photos. Their modified architecture incorporated alterations in the last three convolutional layers, resulting in an enhanced classification accuracy of 84%. One of the drawbacks of the research was the absence of presenting several scenarios to categorize the classes.

Despite extensive studies in the automatic diagnosis and categorization of acute leukemia, certain limitations still exist. Several researchers employed manual feature selection/extraction techniques, necessitating a fundamental understanding of the topic. Furthermore, studies have used deep learning techniques, such as utilizing pre-trained networks, to categorize various forms of leukemia accurately. Nevertheless, their suggested networks lack end-to-end functionality, exhibit computational complexity, and are not operational. Moreover, a notable obstacle in prior research can be attributed to the need for a standardized benchmark database. Many current databases frequently have limited samples and are not easily accessible. The present study aims to address the obstacles above by employing a fusion of graph theory and deep convolutional networks by acquiring an accessible database. 

The contribution of this study can be summarized as follows:Providing a standard database based on two classes, ALL and AML.Presenting an automatic (end-to-end) model for diagnosing acute leukemia using graph theory and deep convolutional networks.Providing the highest level of accuracy when classifying two groups, ALL and AML.

The remaining portion of this article is structured in the following manner:

The second section analyzes the algorithms employed in this investigation. The third section delineates the recommended methodology of this research, encompassing the specifics of data registration, architectural design, and other relevant aspects. The fourth section discusses the simulation findings and compares the current study with algorithms and recent research. Lastly, the fifth section pertains to the conclusion.

## 2. Materials and Methods

This part offers a comprehensive elucidation of generative adversarial networks (GANs) and the use of graph theory in deep CNNs.

### 2.1. General Model of Generative Adversarial Networks

GANs have garnered considerable interest in recent years as a crucial subfield of deep learning. In 2014, J. Goodfellow and colleagues presented these networks [20,21]. GANs are utilized in machine learning to address unsupervised learning tasks. These networks have two models that autonomously detect and recognize patterns in the input data. The two models are commonly referred to as the discriminator and the generator. The discriminator and the generator engage in a competitive process to examine, record, and replicate alterations in the dataset. GANs can generate new samples that are rationally selected from the original dataset. The discriminator is trained using synthetic data generated by the generator. The generator acquires the capacity to produce practical data that may be utilized. Negative training samples refer to the data generated specifically for the discriminator. The generator produces a sample using a random noise vector of a predetermined length as input. The main goal of the generator is to trick the discriminator into correctly labeling its output. The discriminator distinguishes between real data and bogus data generated by the generator. The discriminator has two separate sources of training data. During the training process, the generator generates synthetic samples, which are then classified as negative samples by the discriminator. In contrast, genuine data samples are classified as positive samples.

Mathematically, GANs aim to minimize the following equation during the training phase:(1)$$\begin{array}{l} \log(1-D(G_{\left(Z\right)}))\\ \frac{\min\max}{GD}V(G,D)=E_{x}-P_{data}[\log D(x)]\;\;\;\;\;\;\;\;\;\;\;\;\;\;\;\;\\ +E_{pz(z)}[\log(1-D(G(Z))] \end{array}$$In the above equation, the discriminator (*D*) must be obtained to enable a distinction between genuine and counterfeit data. The equation previously mentioned is unsolvable using a mathematical statement and requires iterative approaches. To mitigate overfitting, the generator function (*G*) is tuned iteratively, with each optimization of function *D* [20,21] occurring once per *k* iterations.

### 2.2. General Model of Graph Convolutional Network

Building models that are capable of data analysis, optimization [22,23], spatial encoding [24], spatial ability [25,26,27], learning content management systems [28,29], prediction, and other tasks is the aim of machine learning [30,31] and its subsets, including federated learning [32,33,34], recurrent neural networks [35], deep learning networks, etc. In this regard, Michael Deferard and his colleagues first introduced the fundamental notion of the graph convolutional network. The researchers utilized signal processing techniques in graph spectral theory for the first time [35]. This enabled the development of convolutional functions and the application of convolutional networks in graph theory. The adjacency and degree matrices hold particular importance in graph theory. An adjacency matrix establishes connections between each vertex in the graph.

Furthermore, the degree matrix can be derived from the adjacency matrix. The diagonal elements of this matrix, which is a diagonal matrix, are equivalent to the total of the edges linking to the corresponding vertex of the matrix. The degree matrix can be denoted as $D\in R^{N\times N}$ and the graph matrix as $W\in R^{N\times N}$, where the i-th diagonal element of the degree matrix is defined as follows [36]:(2)$$D_{ii}=\sum_{i}W_{ij}$$The Laplacian matrix can be alternatively expressed using the following equation:(3)$$L=D-W\in R^{N\times N}$$
(4)$$L=U\Lambda U^{T}$$As stated in the above equation, the Laplacian matrix is formed by subtracting the degree matrices from the adjacency matrix. The matrix is utilized for the computation of graph basis functions. The basis functions of a graph can be derived by applying Singular Value Decomposition (SVD) to the Laplacian matrix. The Laplacian matrix can be defined by considering the matrix of eigenvectors and the matrix of singular values, as expressed in Equation (5). The eigenvectors of the Laplacian matrix are represented by the columns of the eigenvector matrix, as stated in Equation (5). The Fourier transform can be computed using these eigenvectors. Fourier bases can be defined by having diagonal eigenvalues that include $\Lambda =diag([\lambda_{0,\ldots ,}\lambda_{N-1}])$, as expressed by the following relationship:(5)$$U=[u_{0},\ldots ,u_{N-1}]\in R^{N\times N}$$To enhance comprehension, the Fourier transform and inverse Fourier transform of a signal, such as the one depicted, can be precisely specified in Equations (6) and (7) correspondingly:(6)$$\hat{q}=U^{T}q$$
(7)$$q=UU^{T}q=U\hat{q}$$$\hat{q}$, as defined by Equation (6), denotes the Fourier transform of the graph. Furthermore, it is feasible to obtain the feature vector for a signal, denoted as q, by utilizing Fourier bases and the Fourier transform of the graph, as indicated by Equation (7). The graph convolution operator can be computed by convolving two signals in the graph domain using the Fourier transform of each signal. The convolution of two signals, *z* and *y*, using the operator ${*}_{g}$, is defined as the following relationship to enhance comprehension:(8)$$z{*}_{g}=U((U^{T}z)⊙(U^{T}y))$$The equation above uses the g() filter function to define a graph convolution operator with neural networks. Based on the above equation, *z* represents the version filtered by g(L).
(9)$$z{*}_{g}=U((U^{T}z)⊙(U^{T}y))$$Graph convolution can be defined by utilizing the Laplacian matrix and dividing it into singular values and eigenvectors [36,37].
(10)$$\begin{array}{l} y=g(L)z=Ug(\Lambda )U^{T}z\\ =U(g(\Lambda ))⊙(U^{T}z)\\ =U(U^{T}(U_{g}(\Lambda )))⊙(U^{T}z)\\ =z{*}_{g}(Ug(\Lambda )) \end{array}$$

## 3. Proposed Model

This part will comprehensively explain how to register the proposed database, perform data pre-processing, construct a graph, design a network architecture, optimize parameters, and allocate data. This study’s suggested flowchart is graphically depicted in Figure 2.

### 3.1. Data Collection

The dataset included in this work comprised photos of both ALL and AML. These images were obtained from Ghazi Tabriz Medical Sciences Center under the ethical code IR.1401.1.15. The suggested database consisted of 44 patients, 12 males and 32 females, aged 12 to 70 years. Each participant was diagnosed with distinct forms of leukemia, and an oncologist verified their diagnoses. Before data collection, explicit agreement was acquired from all patients to utilize the data gathered in this study. A total of 190 ALL and AML images were obtained from 44 patients who participated in this study. Out of this quantity of photographs, five to seven images could be utilized for each individual.

Typically, there were three sequential procedures required to acquire the planned database. Initially, individuals who were suspected of having leukemia had clinical evaluations and blood tests. If the blood test of the suspect showed any abnormal symptoms, the next stage involved quantifying the number of healthy cells and blast cells in both the peripheral blood smear and bone marrow smear. Once the condition was verified, the advanced-stage oncologist labeled the individual’s blood sample to ascertain the specific type of acute leukemia (ALL or AML). The leukemia classification was determined by visually examining and analyzing the morphological features of lymphocyte and monocyte cells. The data collection process is shown graphically in Figure 3. Figure 4 displays exemplar images of acute leukemia types ALL and AML for one of the individuals. Based on this figure, it is evident that visually distinguishing between different kinds of acute leukemia necessitates specialized expertise, is a time-consuming process, and is susceptible to mistakes.

### 3.2. Pre-Processing

This section explains the pre-processing of the images in the proposed database before entering the proposed architecture. In the first step, because the dimensions of the collected images were not the same, all the images were changed to 226 × 226 and converted to a grayscale output format to reduce the computational volume. As indicated in the data collection section, there was an inequality in the number of class ALL and AML images. This problem can result in a tendency to favor the dominant class in data classification. GAN networks were employed in the second step to address this issue to achieve inter-class balance and data augmentation. To achieve this objective, the generator network was provided with an input of size 1 × 100, which followed a uniform distribution and generated an output of size (226 × 226). This network comprised six convolutional layers with 512, 1024, 2048, 4096, 8192, and 51,076 dimensions. The network utilized the Relu and hyperbolic tangent activation functions in both the hidden and final layers. The *D* network consisted of six completely connected layers that determined the authenticity of the G-generated image, distinguishing between real and fraudulent. The learning rate in this network was 0.001, and the number of iterations was set to 100. After utilizing this network, the number of photos in both classes became equivalent, with the data increasing from 190 to 500. During the third stage, the data were standardized using the Min–Max normalizer [38,39] to simplify the training process. The normalization ensured that the data values were scaled between 0 and 1.

### 3.3. Graph Design

It was necessary to perform a clustering process [40,41,42] on the obtained images to form a graph. To achieve this objective, we obtained a series of superpixels representing distinct regions inside the image. The size of the regions in this study was considered to be 150 for clustering based on trial and error. Subsequently, the mean intensity of pixels inside each extracted region was regarded as the feature vector of each node. Furthermore, the examination of graph edges was conducted by considering the distance and neighborhood of each area. This process resulted in the creation of a graph adjacency matrix, where neighboring places were connected to each other, while non-neighboring areas remained disconnected.

### 3.4. Architecture

This section introduces a dedicated network architecture for acute leukemia diagnosis. After the dropout layer was applied, the input was routed to the six graph convolutional layers enabled by the Relu activation function. Following batch normalization, the data were passed through a dropout layer to prevent overfitting. Finally, the output was treated as a flattening layer divided into ALL and AML classes, using a fully connected layer and the Sofmax. Each node in the constructed graph had a sample because its feature vector represented the average pixel intensity in each region. The input dimension of the graph convolution layer was assumed to be 32. The second layer generated a pi-node graph, with 32 samples per vertex. This operation continued up to the sixth layer and caused the sixth layer to create an *A*-node graph with dimensions of two. A vector of elements was created, with two samples at each node. The samples were then divided into ALL and AML, based on the number of points scored, using the softmax function. In this context, *P*_1–6_ also denoted the coefficients of Cheby Sheff polynomials, whose value was assumed to be 1 through trial and error. The method of forming a graph in the proposed architecture is shown in Figure 5. Also, the details of the different layers are clear in Figure 6. The number of layers and their dimensions are shown in Table 1.

### 3.5. Training, Validation, and Test Series

The hyperparameters used in the proposed model were organized using the trial and error method. Table 2 shows the tested parameters for various network sections and the optimal parameters. Thus, we attempted to consider the most influential parameter in the proposed model.

To assess the network’s performance, a random selection was made, allocating 70% of the data from the dataset to the training set, 20% to the validation set, and 10% to the test set. We also used 5-fold cross-validation to evaluate the data. This criterion ensured that all the data would be included in both the training and test processes. Figure 7 depicts the 5-fold cross-validation process.

## 4. Results

The results of the proposed model will be presented in this section. The proposed architecture was implemented using the Python programming language, while the data preparation simulations were conducted in the MATLAB 2019a environment. In addition, the findings were produced by the Google Colab Premium edition, which was equipped with a GPU t60 and 64 GB of RAM.

Two subsections make up this section. To visually demonstrate that the architecture considered for the current application was in an ideal state, the optimization findings for the network architecture are displayed in the first section. The outcomes of the suggested model for the automated detection of lie detectors are shown in the second subsection.

### 4.1. Optimization Results

The proposed network architecture was structured according to a process of experimentation, as elucidated in the third part. In the suggested model for the automatic categorization of acute leukemia, we made efforts to carefully evaluate the most optimal architecture based on speed and accuracy criteria. In this subsection, we will visually demonstrate the effectiveness of the proposed architecture.

To construct the proposed architectural framework, we considered the number of distinct layers and assessed the model based on its speed and correctness. The outcomes acquired for selecting the number of layers are displayed in Figure 8. Based on the same data, it was evident that opting for six convolutional graph layers was a cost-effective choice in terms of both speed and accuracy. The chosen numbers for the Chebyshev polynomial were also regarded as variables. Figure 9 displays the accuracy results of the network for various selected numbers of Chebi Sheff. Based on the given figure, it is evident that selecting *P*_1_ − *P*_6_ = 1 resulted in a 99% accuracy rate for the suggested model. As per Section 3, we employed the clustering technique to ascertain the dimensions of the regions. This approach utilized a heuristic method for graph embedding, involving iterative experimentation and refinement. Figure 10 depicts the separate diameters of the regions for two samples, ALL and AML. Table 3 presents the accuracy of the recommended model based on different aspects of the regions. The table demonstrates that the choice of 100 regions for the proposed model proved successful.

### 4.2. Simulation Results

This subsection will present the outcomes derived from the suggested model. Figure 11 displays the accuracy and error of the proposed deep network during the training and validation of the model across 150 iterations. Based on the same data, it was evident that the model successfully completed the learning process. Furthermore, the model’s validation confirmed that it attained stability after 120 iterations and obtained an accuracy of 99%. Furthermore, the model error dropped as the number of repeats increased and eventually achieved its lowest level. Table 4 displays the results obtained for classifying acute leukemia into ALL and AML classes, using several evaluation criteria such as accuracy, sensitivity, precision, specificity, and kappa coefficient. The binary classification of ALL and AML exceeded 95% for all evaluation indicators. Figure 12 depicts the classification results of ALL and AML classes using the 5-fold cross-validation criterion. According to this figure, the classification results in different folds were greater than 95%, indicating that overfitting did not occur during network training. Figure 13 displays the confusion matrix and statistical analysis of the receiver operating characteristic (ROC) for categorizing acute leukemia into two distinct classes, ALL and AML. Based on the scatter matrix, it was evident that the suggested model misclassified only two samples belonging to class AML. Furthermore, the statistical analysis demonstrated that the graph fell within the permitted range of values (0.9–1), thus confirming the accuracy of the binary classification and indicating that overfitting was not present during the training of the model. Figure 14 displays the T-SNE graph for both the raw data and the data that were treated in the last layer. Based on the diagram, it is evident that the samples were distinctly segregated from each other according to the designed model. The available information indicated that the suggested model for the automatic diagnosis of ALL and AML leukemia demonstrated high reliability and promising performance.

## 5. Discussion 

In this section, the proposed deep model, based on the combination of graph theory and convolutional networks, will be examined in relation to other recent studies and algorithms used to classify acute leukemia. 

The performance results related to recent studies, the method used, and the proposed method are presented in Table 5. According to Table 5, as can be seen, the proposed method achieved the highest performance compared to previous studies. So, the binary classification accuracy for the proposed model was 99.4. However, a one-to-one comparison with prior studies is unfair due to the different databases. Future studies can evaluate their proposed models using the database collected in this study (which is open-access). To make the comparison with previous studies fair, we used recent methods such as CNN, ResNet60, and VGG 16 compared to our model. These methods have been widely used in recent studies. Accordingly, the proposed database was used to train the CNN, ResNet60, and VGG 16 networks. Also, the proposed architecture without graph layers was considered for the CNN architecture. The performance obtained in 150 iterations is shown in Figure 15. Accordingly, as it is known, the proposed network performed at the best level compared to other networks, which indicated the optimal architecture.

The acquired images could contain ambient noise. Therefore, it was imperative to assess the suggested model in noisy settings. To achieve this objective, we intentionally added white Gaussian noise to the images at various signal-to-noise ratios (SNRs) and assessed the effectiveness of the suggested model in this noisy setting. Figure 16 displays the noise introduced to the photographs at various SNRs. Furthermore, Figure 17 illustrates the performance of the suggested model compared to previous models. Based on Figure 17, it is evident that the proposed network effectively preserved its resilience against ambient noise. The suggested model’s ability to withstand external influences was advantageous when utilizing convolutional graph networks and their distinctive architecture.

Utilizing the approach outlined in this paper, employing transfer learning and integrating filters emerges as the most effective strategy based on current state-of-the-art methodologies [45]. Utilizing the machine learning approach outlined in this paper has yielded superior outcomes, emphasizing its efficacy in optimizing results and fortifying the model’s resilience and adaptability across different platforms [46]. Although the proposed model performed well, this study, like others, has drawbacks. One of the drawbacks of this research is the use of binary categorization. In future studies, the number of classes can be expanded to include more classes such as ALL, AML, and so on, in addition to the existing classes ALL and AML. Furthermore, it is feasible to evaluate the efficacy of data augmentation by comparing the performance of traditional algorithms, such as rotation and shift, with GANs.

## 6. Conclusions

This study introduced an advanced end-to-end approach for the automated diagnosis of acute leukemia classes ALL and AML. The end-to-end deep model consisted of a fusion of graph theory and CNNs. It included six graph Conv layers and a Softmax layer in the output for the computation of AML and ALL scores. This work involved the collection of a standard database consisting of images of blood samples from 44 individuals, together with their corresponding labels ALL and AML. The evaluation metrics, namely, accuracy and kappa coefficient, were reported as 99% and 0.85%, respectively. Furthermore, the proposed model was tested in a noisy environment and demonstrated the ability to maintain an accuracy rate greater than 90% for categorizing two classes, ALL and AML, across a wide range of SNRs. Given the model’s outstanding performance, it can serve as a valuable tool for oncologists to classify different leukemia types accurately.

## Figures and Tables

**Figure 1 bioengineering-11-00644-f001:**
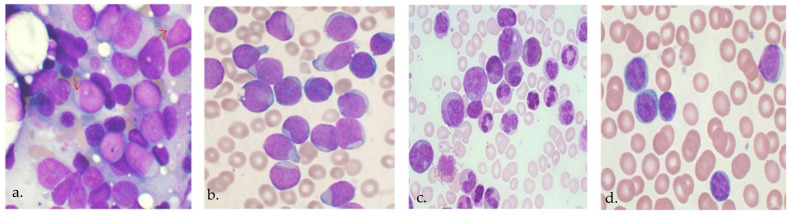
Different types of acute leukemia, including (**a**) AML, (**b**) ALL, (**c**) CML, and (**d**) CLL.

**Figure 2 bioengineering-11-00644-f002:**
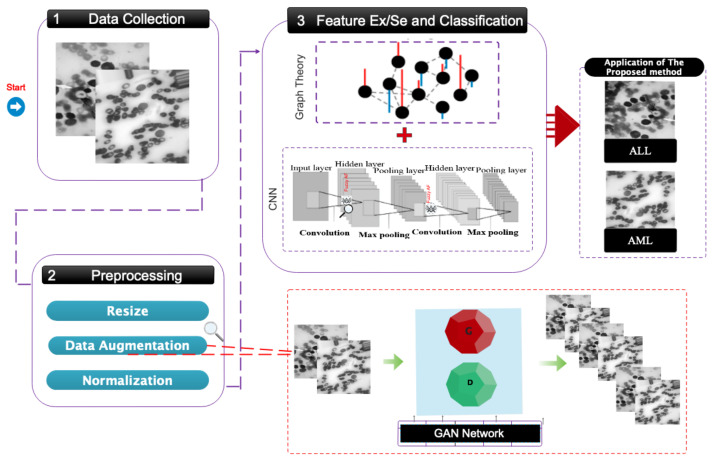
The proposed primary framework for the automated diagnosis of acute leukemia involves categorizing it into two classifications: ALL and AML.

**Figure 3 bioengineering-11-00644-f003:**
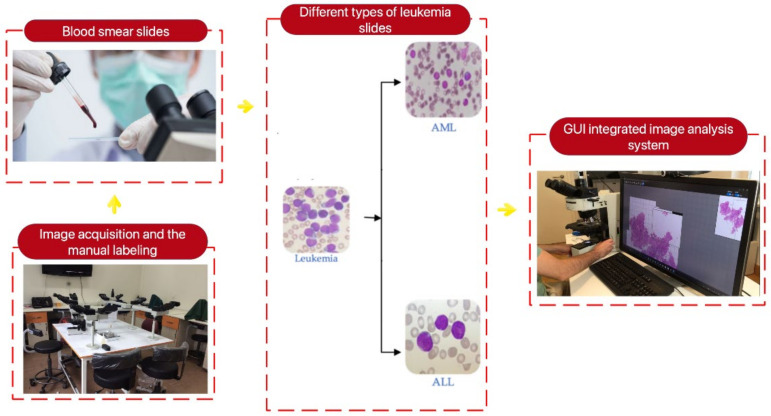
The data collection process for ALL and AML classes.

**Figure 4 bioengineering-11-00644-f004:**
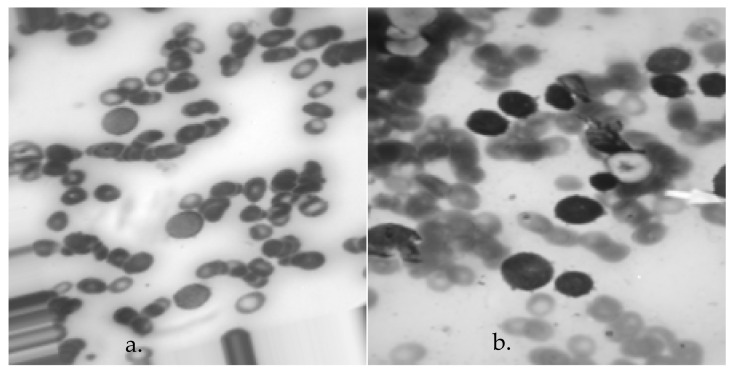
An example of the images taken in the proposed database for (**a**) ALL and (**b**) AML.

**Figure 5 bioengineering-11-00644-f005:**
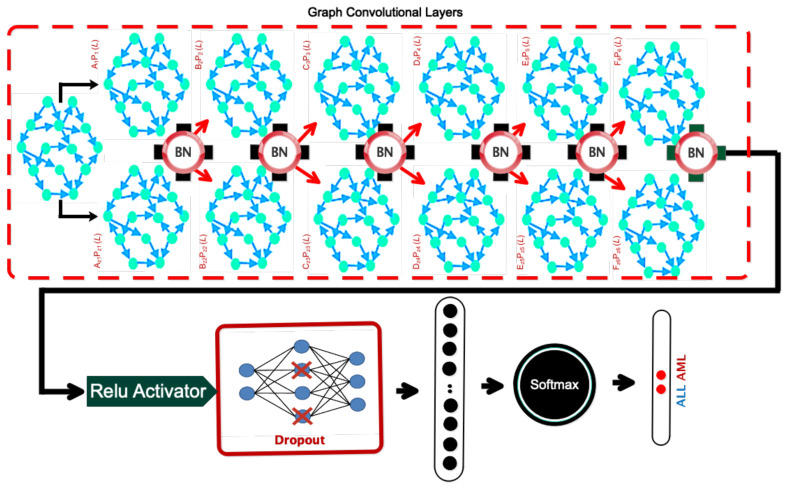
Graphic view of the proposed architecture.

**Figure 6 bioengineering-11-00644-f006:**
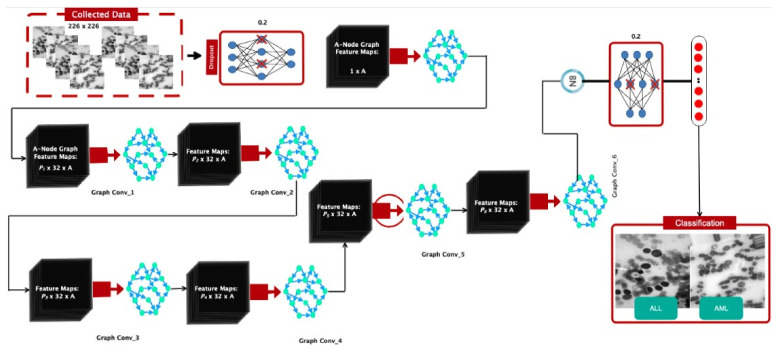
Details of the layers in deeply organized architecture.

**Figure 7 bioengineering-11-00644-f007:**
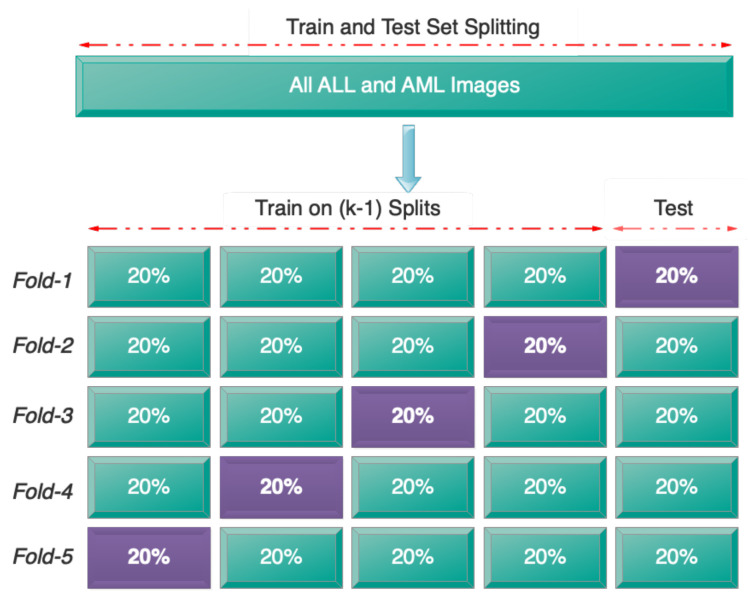
Five-fold cross-validation operation.

**Figure 8 bioengineering-11-00644-f008:**
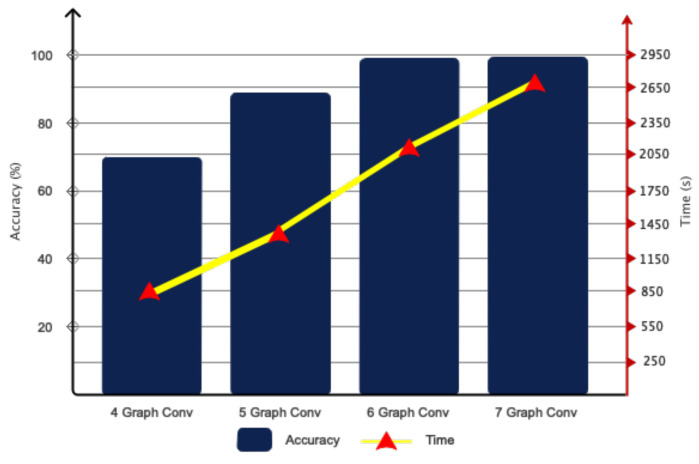
The graph convolutional architecture was tested with various polynomial coefficients.

**Figure 9 bioengineering-11-00644-f009:**
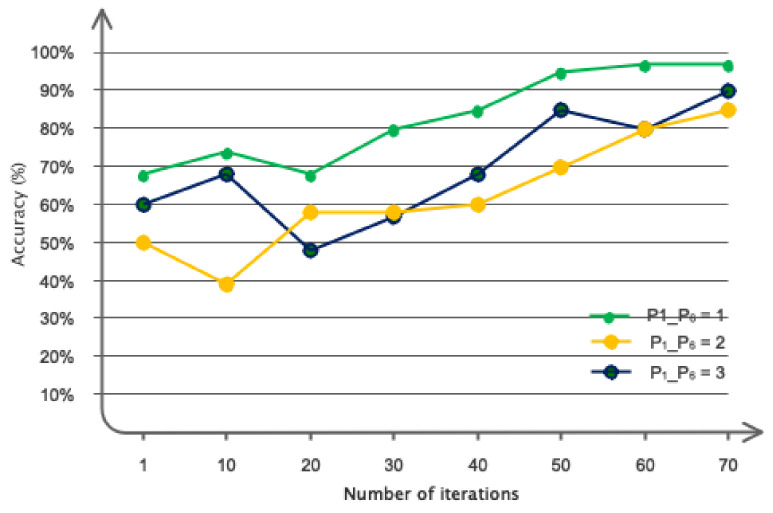
Different polynomial coefficients were examined in the graph convolutional architecture.

**Figure 10 bioengineering-11-00644-f010:**
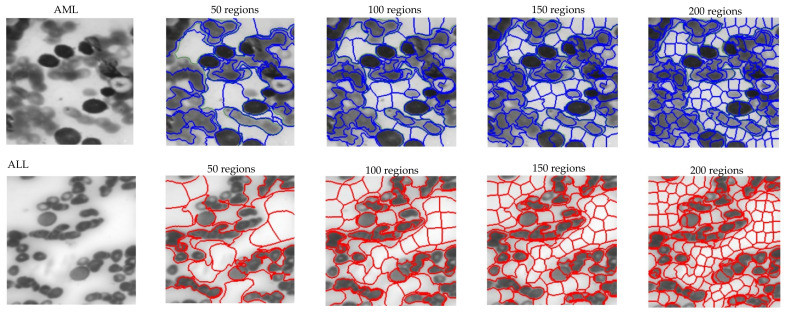
Specific locations were chosen for the ALL and AML samples for graph embedding.

**Figure 11 bioengineering-11-00644-f011:**
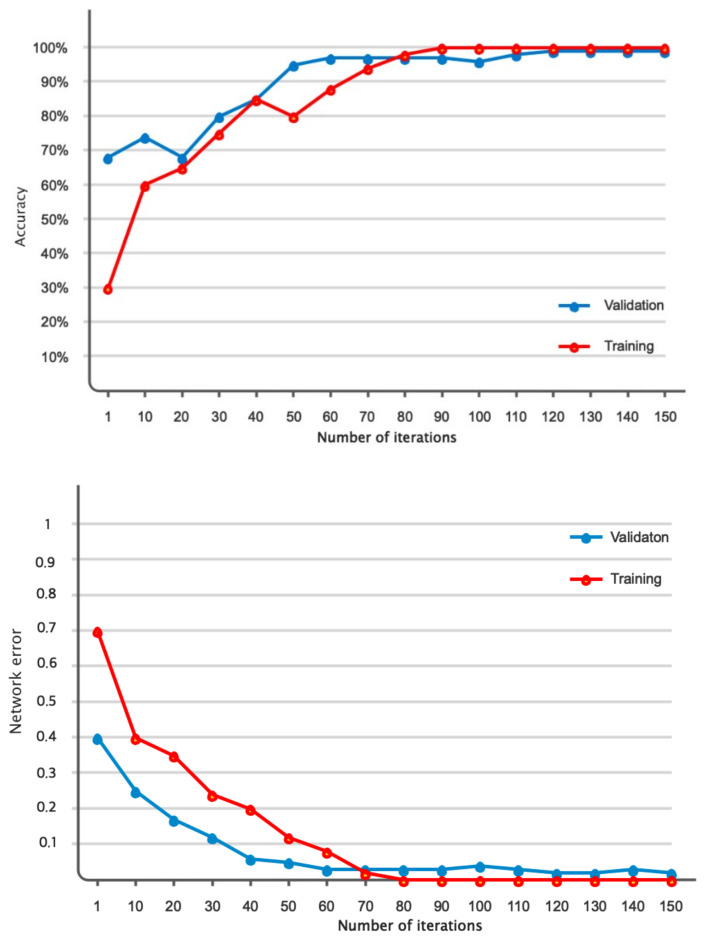
The suggested model’s accuracy and error for training and validation assessed across 150 iterations.

**Figure 12 bioengineering-11-00644-f012:**
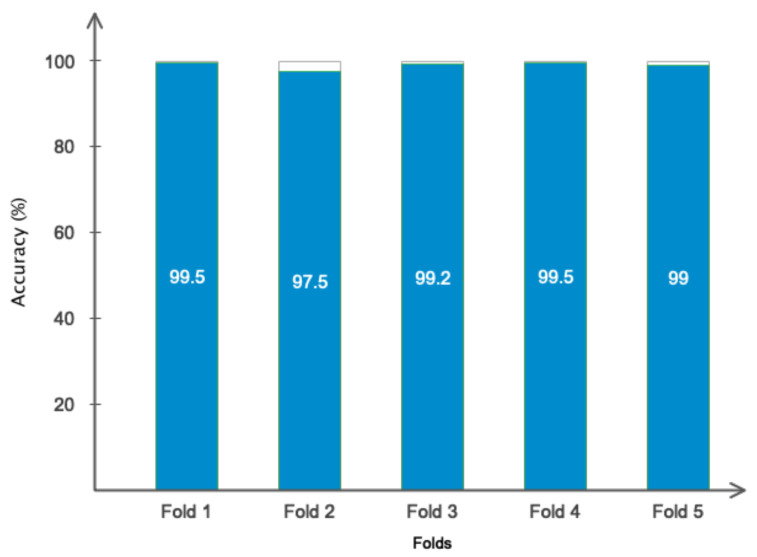
The classification accuracy results based on 5-fold cross-validation criteria.

**Figure 13 bioengineering-11-00644-f013:**
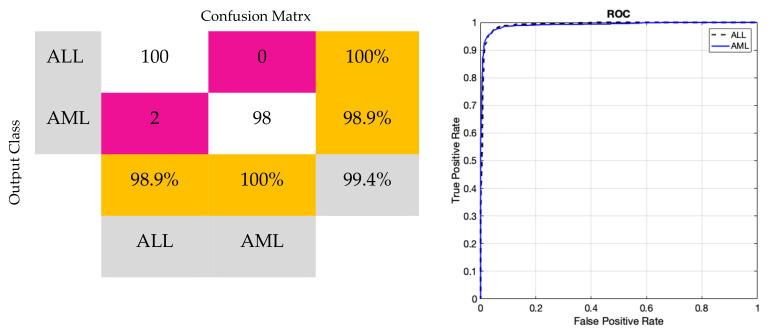
Performance of confusion matrix and ROC curve in the proposed model.

**Figure 14 bioengineering-11-00644-f014:**
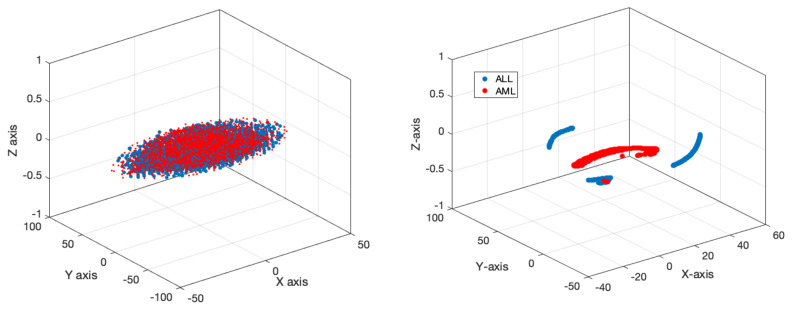
Examples of two categories of veracity and falsity for unprocessed data and the fully connected network layer.

**Figure 15 bioengineering-11-00644-f015:**
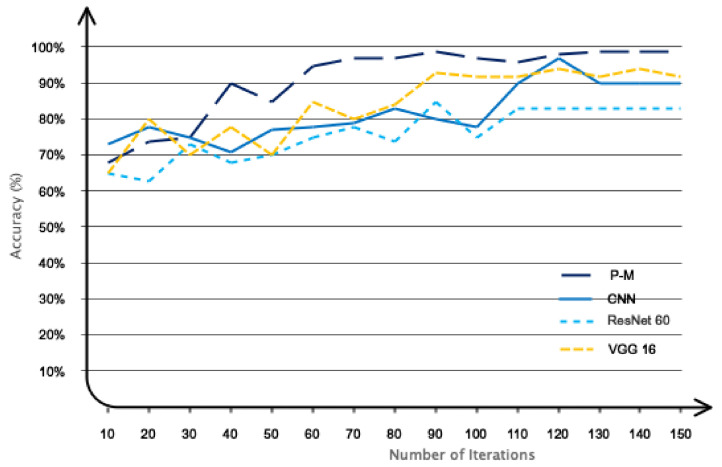
The performance of the proposed network compared to other networks.

**Figure 16 bioengineering-11-00644-f016:**
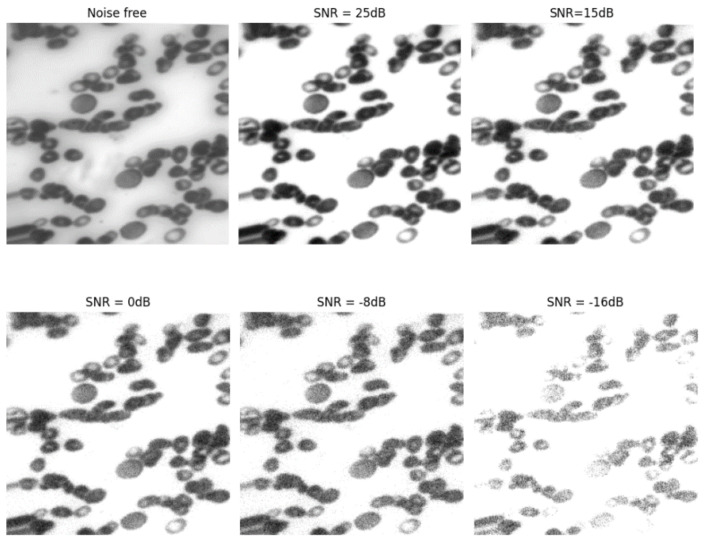
A sample of images with varying decibels of noise applied.

**Figure 17 bioengineering-11-00644-f017:**
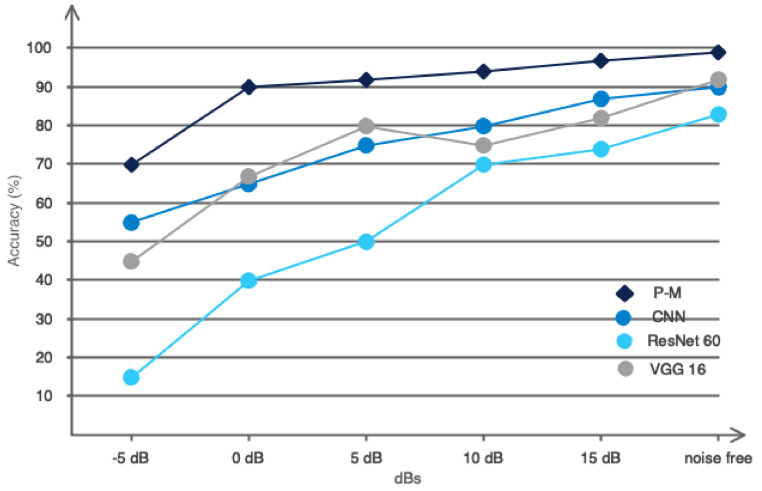
The performance of the proposed network in relation to other networks.

**Table 1 bioengineering-11-00644-t001:** Layers, weight, bias, and parameters in the proposed architecture.

Layer	Shape of Weight Tensor	Shape of Bias	Number of Parameters
**Graph Conv1**	(*P*_1_, 32, 32)	32	1024 × *P*_1_ + 32
**Batch Norm**	(32)	32	64
**Graph Conv2**	(*P*_2_, 32, 32)	32	1024 × *P*_2_ + 32
**Batch Norm**	(32)	32	64
**Graph Conv3**	(*P*_3_, 32, 32)	32	1024 × *P*_3_ + 32
**Batch Norm**	(32)	32	64
**Graph Conv4**	(*P*_4_, 32, 32)	32	1024 × *P*_4_ + 32
**Batch Norm**	(32)	32	64
**Graph Conv5**	(*P*_5_, 32, 32)	32	1024 × *P*_5_ + 32
**Batch Norm**	(32)	32	64
**Graph Conv6**	(*P*_6_, 32, 2)	2	64 × *P*_6_ + 32
**Batch Norm**	(16)	16	32
**Softmax**	-	2	2 × *A* × *P*_6_

**Table 2 bioengineering-11-00644-t002:** Choice of suggested network architecture’s ideal parameters.

Parameters	Values	Optimal Value
Batch Size in GAN	4, 6, 8, 10, 12	**12**
Optimizer in GAN	Adam, SGD, Adamax	**Adamax**
Number of CNN Layers	3, 4, 5, 6	**6**
Learning Rate in GAN	0.1, 0.01, 0.001, 0.0001	**0.001**
Number of Graph Conv Layers	2, 3, 4, 5, 6, 7	**6**
Batch Size in GCN	8, 16, 32	**32**
Batch normalization	Relu, Leaky-Relu	**Relu**
Learning Rate in GCN	0.1, 0.01, 0.001, 0.0001, 0.00001	**0.0001**
Dropout Rate	0.1, 0.2, 0.3	**0.2**
Weight of optimizer	$4\times 10^{-3},4\times 10^{-4},4\times 10^{-5},4\times 10^{-6},4\times 10^{-7}$	$4\times 10^{-4}$
Error function	MSE, Cross Entropy	**Cross Entropy**
Optimizer in GCN	Adam, SGD, Adadelta, Adamax	**SGD**

**Table 3 bioengineering-11-00644-t003:** Performance of the proposed model by changing the clustering regions.

**Regions**	50	**100**	150	200
**Accuracy**	94.1%	**99.4%**	91%	82%

**Table 4 bioengineering-11-00644-t004:** The performance of the proposed network assessed using various assessment indices.

Measurement Index	Performance (%)
**Accuracy**	99.4
**Sensitivity**	99.2
**Precision**	98.1
**Specificity**	97.3
**Kappa coefficient**	0.85

**Table 5 bioengineering-11-00644-t005:** Evaluating the efficacy of the proposed approach against recent research findings.

Ref.	Dataset	Classification	Methods	Accuracy
**Zhou et al.** [12]	ALL-IDB1	ALL	FCNN	85%
**Khandkar et al.** [13]	ALL-IDB1 and CNMC 2019	ALL, AML	Thresholding	95%
**Chola et al.** [14]	HPBC	Leukemia types	BCNet	98.51%
**Bhute et al.** [15]	Private dataset	Leukemia	Pre-trained networks (VGG16, Resnet60, Inception V3)	90%
**Rastogi et al.** [16]	ALL-IDB2	ALL-AML	Leufeatx	96.15%
**Dese et al.** [17]	Private dataset	Leukemia types	Deep learning methods	95%
**Ansari et al.** [18]	Private dataset	ALL-AML	Type 2 fuzzy + CNN	98%
**Abhishek et al.** [19]	Private dataset	Leukemia types	VGG 16	85%
**Areen et al.** [43]	ALL-IDB	Leukemia types	Pre-trained networks (VGG16, Resnet60, Inception V3)	94%
**Awais et al.** [44]	Private dataset	ALL	CNNs	99.15%
**Proposed method**	New dataset (ALL + AML)	ALL-AML	Graph theory + CNN	99.4%

## Data Availability

The data are private and the University Ethics Committee does not allow public access to the data.

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
