# Peer review of "Automatic Detection of Acute Leukemia (ALL and AML) Utilizing Customized Deep Graph Convolutional Neural Networks"

_bioengineering, 2024, doi:10.3390/bioengineering11070644_

Round 1
Reviewer 1 Report
Comments and Suggestions for Authors
I examined your work titled "Automatic detection of acute leukemia (ALL and AML) utilizing customized deep graph convolutional neural networks" in detail. I listed the deficiencies I saw in the study in items. In the summary section, it is stated that 670 images of 44 patients were used. At this stage, brief information about how these images were obtained should be given. While four different classes are mentioned in Figure 1, why were the results taken from 2 classes? The contribution section should highlight the study's contributions to the literature. How many images were used when obtaining the test results, do these images include images obtained by data multiplexing, and these image numbers should be given in confusion matrices. Why were GAN architectures used in the study? Wouldn't working with more original data instead of image multiplexing be more excellent? The data sets used in the literature contain a large number of images. I recommend testing your model on one of these datasets. It can be seen that the proposed model section is written very clearly, and Figure 2 is not explained in detail. Especially since the number of data is small, it needs to be proven how overfitting is prevented after the multiplexing step and whether the proposed model is successful in other leukemia data sets.
Comments on the Quality of English LanguageSpelling and grammatical errors need to be reviewed.
Author Response
Reviewer#1:
Comments:
Examined your work titled "Automatic detection of acute leukemia (ALL and AML) utilizing customized deep graph convolutional neural networks" in detail. I listed the deficiencies I saw in the study in items.
- ⎫ While thanking the esteemed reviewer for a thorough review of the manuscript version. We, the authors of the article, believe that your suggestions have been very useful and effective in improving the scientific version of the manuscript. We carefully answered all the questions and suggestions of the esteemed reviewer and added them to the manuscript version.
- 1. In the summary section, it is stated that 670 images of 44 patients were used. At this stage, brief information about how these images were obtained should be given.
- ⎫ The manuscript is revised based on this comment. According to the reviewer, the method of data collection is fully displayed in Figure 3.
Figure 3. Data collection process for ALL and AML classes.
- 2. While four different classes are mentioned in Figure 1, why were the results taken from 2 classes?
- ⎫ We have described different types of leukemia in the introduction. However, our focus in the proposed method, such as studies [*, **], has only been on the classification of widely used ALL and AML classes. These classes are very useful for diagnosing leukemia and the high accuracy plays an essential role in their classification.
[*] Brunning, Richard D. "Classification of acute leukemias." Seminars in diagnostic pathology. Vol. 20. No. 3. WB Saunders, 2003.
[**] Casasnovas, René Olivier, et al. "Immunological classification of acute myeloblastic leukemias: relevance to patient outcome." Leu
- 3. The contribution section should highlight the study's contributions to the literature.
- ⎫ The manuscript is revised based on this comment. According to the reviewer, the contribution of the article is as follows:
1. Providing a standard database based on two classes, ALL and AML.
2. Presenting an automatic (End-to-End) model for diagnosing acute leukemia using graph theory and deep convolutional networks.
3. Providing the highest level of accuracy when classifying two groups, ALL and AML.
- 4. How many images were used when obtaining the test results, do these images include images obtained by data multiplexing, and these image numbers should be given in confusion matrices.
- ⎫ The manuscript is revised based on this comment. The number of validation data for evaluating the proposed architecture is 100 samples, which is presented in Figure 11. Also, the total number of available samples is 190 samples, which has reached 500 samples after data augmentation using GAN. 70% of this value is used for training data, 20% for validation and 10% for testing.
- 5. Why were GAN architectures used in the study?
- ⎫ We have used GAN networks to artificially increase the dimensions of the database. By increasing the deep architecture database, a better trained proposal and the occurrence of over-fitting phenomenon is minimized in it. For this purpose, we have considered a proposed architecture for GAN networks:
“To address this issue, GAN networks have been employed in the second step to achieve inter-class balance and data augmentation. To achieve this objective, the generator network is provided with an input of size 1 x 100, which follows a uniform distribution, and generates an output of size (226 x 226). This network comprises six convolutional layers with dimensions of 512, 1024, 2048, 4096, 8192, and 51076. The network has utilized the Relu and hyperbolic tangent activation functions in both the hidden layers and the final layer. The D network consists of 6 completely connected layers that determine the authenticity of the image generated by G, distinguishing between real and fraudulent. The learning rate in this network is 0.001 and the number of iterations is set to 100. After utilizing this network, the quantity of photos in both classes has become equivalent, with the data increasing from 190 to 500.”
- 6. Wouldn't working with more original data instead of image multiplexing be more excellent?
- ⎫ As you know, deep learning networks are big data networks and need a lot of data for training. For this reason, in this research, adversarial generative networks have been used after collecting data in order to increase the data so that the network training process is done well. Adversarial generative networks have been widely used to increase data in recent researches [*, **, ***] and have shown very promising results.
[*] Ardabili, S. Z., Bahmani, S., Lahijan, L. Z., Khaleghi, N., Sheykhivand, S., & Danishvar, S. (2024). A novel approach for automatic detection of driver fatigue using EEG signals based on graph convolutional networks. Sensors, 24(2), 364.
[**] Khaleghi, Nastaran, et al. "Visual saliency and image reconstruction from EEG signals via an effective geometric deep network-based generative adversarial network." Electronics 11.21 (2022): 3637.
[***] Ansari, Sanam, et al. "A customized efficient deep learning model for the diagnosis of acute leukemia cells based on lymphocyte and monocyte images." Electronics 12.2 (2023): 322.
- 7. The data sets used in the literature contain a large number of images. I recommend testing your model on one of these datasets.
- ⎫ The databases that are publicly available are all 4-class databases. However, the focus of this study is on the classification of 2 classes. For this reason, it is not possible to use another database. There are many studies [12, 13, 18, 34] that have used two-class private database in order to evaluate their algorithm.
- 8. It can be seen that the proposed model section is written very clearly, and Figure 2 is not explained in detail. Especially since the number of data is small, it needs to be proven how overfitting is prevented after the multiplexing step and whether the proposed model is successful in other leukemia data sets.
- ⎫ The manuscript has been revised based on this feedback. To demonstrate the proposed model's optimal efficiency and the absence of overfitting, we used 5-fold cross validation. Using this criterion ensures that all data is present during both the training and testing processes. Figure 7 depicts the 5-fold cross validation process graphically.
Figure 5. 5-fold cross validation operation.
Figure 12 depicts the classification results of ALL and AML classes using the 5-fold cross validation criterion. According to this figure, the classification results in different folds are greater than 95%, indicating that overfitting did not occur during network training.
Figure 12. The classification accuracy results are based on 5-fold cross validation criteria.

Reviewer 2 Report
Comments and Suggestions for Authors
See the attachment.

Pass
Author Response
Reviewer#2:
Comments:
In this paper, the automatic detection of acute leukemia (ALL and AML) utilizingcustomized deep graph convolutional neural networks was studied. Fromtheexperiment, it seems that the proposed method achieved high accuracy. After readingthe whole paper, I have the following comments:
- ⎫ While thanking the esteemed reviewer for a thorough review of the manuscript version. We, the authors of the article, believe that your suggestions have been very useful and effective in improving the scientific version of the manuscript. We carefully answered all the questions and suggestions of the esteemed reviewer and added them to the manuscript version.
- 1. The languages and sentences should be well polished, especially those grammar errors.
- ⎫ The manuscript is revised based on this comment. According to the reviewer's opinion, we have re-checked the manuscript in terms of spelling and grammar, and we have corrected all spelling and writing errors.
- 2. The CNN is a quite commnon learning model. Compared with some advanced models in IEEE Transactions on Industrial Electronics, vol. 69, no. 12, pp. 13462-13472, 2022. IEEE/ASME Transactions on Mechatronics, vol. 28, no. 5, pp. 2645-2656, 2023, what is your advantage? More discussions should be made.
- ⎫ The manuscript is revised based on this comment. With respect to the opinion of the respected reviewer, CNN was used in the mentioned study for fault detection, but the deep model proposed in this study, which is a combination of convolutional graph network with type 2 fuzzy, is used to classify two classes of medical ALL and AML. However, according to the opinion of the respected referee, the relevant research has been cited as a reference [22].
- 3. The number of mathematical equation should be presented in the right side.
- ⎫ The manuscript is revised based on this comment. Thanks to the opinion of the respected referee, the numbers of the mathematical equation are on the left side in the MDPA publication format itself. However, according to the reviewer's opinion, the number of mathematical equations has been added on the right side.
- 4. The earliest comparison paper was published in 2022, which is too old. Some more recent papers, see, e.g., those papers published in 2024 should be added for comparison.
- ⎫ The manuscript is revised based on this comment. With respect to the opinion of the respected referee, the articles 15, 18, 19, which have been compared with the proposed method, are related to the years 2023 and 2024. However, in order to respect the opinion of the respected referee, two 2024 articles [*,**] have been added to Table 5 for comparison.
[*] Al-Bashir, Areen K., Ruba E. Khnouf, and Lamis R. Bany Issa. "Leukemia classification using different CNN-based algorithms-comparative study." Neural Computing and Applications (2024): 1-16.
[**] Awais, Muhammad, et al. "ALL classification using neural ensemble and memetic deep feature optimization." Frontiers in Artificial Intelligence 7 (2024): 1351942.
- 5. For experiment, what happens when the SINR is smaller than 0? any solutions?
- ⎫ The manuscript is revised based on this comment. According to Figure 8, as can be seen in SNR=-5, the classification accuracy of the proposed model and the compared models has decreased more steeply, and for the proposed model, the classification accuracy has reached its lowest value, i.e. 70%.
- 6. Reference format should be unified.
- ⎫ All reference formats have been modified and changed to ACS format according to MDPA publication guidelines.

Reviewer 3 Report
Comments and Suggestions for Authors
This paper looks interesting, but some points need to be revised:
- Lines 106-115. It is not clear what is the purpose of this paper. Please revise this part.
- The manuscript has no discussion section. Can you explain why? Probably this section can be useful to compare previous papers.
- Lines 39-40: "This disease has the ability to spread to other organs, including the spleen, brain, liver, and kidney, by traveling through the circulation". The authors should consider also recent papers like these: -- doi: 10.3390/surgeries5020018 --- doi: 10.1001/jama.2017.0693 -- doi: 10.1007/s00520-019-05197-y
- Lines 399-401: "Furthermore, the suggested model... across a diverse array of SNRs." This part should be revised. What do the authors mean?
- Lines 85-86: "A drawback of this research can be attributed to 85 the limited number of classification classes." This is related to what? Reference
Comments on the Quality of English LanguageMinor editing of English language required
Author Response
Reviewer#3:
Comments:
This paper looks interesting, but some points need to be revised:
- ⎫ While thanking the esteemed reviewer for a thorough review of the manuscript version. We, the authors of the article, believe that your suggestions have been very useful and effective in improving the scientific version of the manuscript. We carefully answered all the questions and suggestions of the esteemed reviewer and added them to the manuscript version.
- 1. - Lines 106-115. It is not clear what is the purpose of this paper. Please revise this part.
- ⎫ The manuscript is revised based on this comment. Yes, the opinion of the honorable judge is absolutely correct. Based on the opinion of the respected referee, the article has been modified as follows:
1. Providing a standard database based on two classes, ALL and AML.
2. Presenting an automatic (End-to-End) model for diagnosing acute leukemia using graph theory and deep convolutional networks.
3. Providing the highest level of accuracy when classifying two groups, ALL and AML.
- 2. - The manuscript has no discussion section. Can you explain why? Probably this section can be useful to compare previous papers.
- ⎫ The manuscript is revised based on this comment. Based on the reviewer's opinion, the discussion section has been added to the manuscript as follows:
“5. Discussion
In this section, the proposed deep model based on the combination of graph theory and convolutional networks with other recent studies and algorithms used to classify acute leukemia will be examined.
The performance results related to recent studies along with the method used in them along with the proposed method are presented in Table 5. According to Table 5, as can be seen, the proposed method has been able to achieve the highest performance compared to previous studies; So, the binary classification accuracy for the proposed model is 99.4. However, due to the different databases, a one-to-one comparison with previous studies is not fair. Future studies can evaluate their proposed models using the database collected in this study (which is open access). In order to make the comparison with previous studies fair, we have used recent methods such as CNN [12], ResNet60 [15] and VGG 16 [19] compared to our model. These methods have been widely used in recent studies. Accordingly, the proposed database was used to train CNN, ResNet60 and VGG 16 networks. Also, for the CNN architecture, the proposed architecture without graph layers was considered. The performance obtained in 150 iterations is shown in Figure 12. Accordingly, as it is known, the proposed network has been able to have the best performance compared to other networks, which indicates the optimal architecture.
Table 5. Evaluating the efficacy of the proposed approach in relation to recent research findings.
|
Ref. |
Dataset |
Classification |
Methods |
Accuracy |
|
Zhou et al. [12] |
ALL-IDB1 |
ALL |
FCNN |
85% |
|
Khandkar et al. [13] |
ALL-IDB1 and CNMC 2019 |
ALL, AML |
Thresholding |
95% |
|
Chola et al. [14] |
HPBC |
Leukemia types |
BCNet |
98.51% |
|
Bhute et al. [15] |
Private dataset |
Leukemia |
Pre-trained networks (VGG16, Resnet60, Inception V3) |
90% |
|
Rastogi et al. [16] |
ALL-IDB2 |
ALL-AML |
Leufeatx |
96.15% |
|
Dese et al. [17] |
Private dataset |
Leukemia types |
Deep Learning Methods |
95% |
|
Ansari et al. [18] |
Private dataset |
ALL-AML |
Type 2 fuzzy + CNN |
98% |
|
Abhishek et al. [19] |
Private dataset |
Leukemia types |
VGG 16 |
85% |
|
Areen et al. [33] |
ALL-IDB |
Leukemia types |
Pre-trained networks (VGG16, Resnet60, Incep-tion V3) |
94% |
|
Awais et al. [34] |
Private dataset |
ALL |
CNNs |
99.15% |
|
Proposed method |
New dataset (ALL+ AML) |
ALL-AML |
Graph theory + CNN |
99.4% |
Figure 12. The performance of the proposed network in relation to other networks.
It is evident that the acquired images may contain ambient noise. Therefore, it is imperative to assess the suggested model in noisy settings. To achieve this objective, we intentionally added white Gaussian noise to the images at various signal-to-noise ratios (SNRs) and assessed the effectiveness of the suggested model in this noisy setting. The Figure 13 displays the noise that was introduced to the photographs at various SNRs. Furthermore, Figure 14 illustrates the performance of the suggested model in comparison to previous models. Based on Figure 14, it is evident that the proposed network has effectively preserved its resilience against ambient noise. The suggested model's ability to withstand external influences is advantageous when utilizing convolutional graph networks and its distinctive architecture.
|
|
|
|
Figure 13. A sampling of images with varying decibels of noise applied.
Figure 14. The performance of the proposed network in relation to other networks.
Although the proposed model performed well, this study, like others, has drawbacks. One of the drawbacks of the research is the use of binary categorization. In future studies, the number of classes can be expanded to include more classes such as ALL, AML, and so on, in addition to the existing classes ALL and AML. Furthermore, it is feasible to evaluate the efficacy of data augmentation by comparing the performance of traditional algorithms, such as rotation and shift, with GAN networks.”
- 3. - Lines 39-40: "This disease has the ability to spread to other organs, including the spleen, brain, liver, and kidney, by traveling through the circulation". The authors should consider also recent papers like these: -- doi: 10.3390/surgeries5020018 --- doi: 10.1001/jama.2017.0693 -- doi: 10.1007/s00520-019-05197-y
- ⎫ The manuscript is revised based on this comment. According to the opinion of the respected referee, the relevant studies have been very suitable for improving the quality of the structure of the article, which have been added to the article as references [6], [7] and [8].
- 4. - Lines 399-401: "Furthermore, the suggested model... across a diverse array of SNRs." This part should be revised. What do the authors mean?
- ⎫ The manuscript is revised based on this comment. Yes, it is absolutely true. Based on the opinion of the respected judge, the relevant sentence was modified as follows:
“Furthermore, the proposed model was tested in a noisy environment and demonstrated the ability to maintain an accuracy rate greater than 90% for the categorization of two classes, ALL and AML, across a wide range of SNRs.”
- 5. - Lines 85-86: "A drawback of this research can be attributed to 85 the limited number of classification classes." This is related to what? Reference
- ⎫ The manuscript is revised based on this comment. Yes, it is absolutely true. Based on the opinion of the respected judge, the relevant sentence was modified as follows:
“Although the proposed model performed well, this study, like others, has drawbacks. One of the drawbacks of the research is the use of binary categorization. In future studies, the number of classes can be expanded to include more classes such as ALL, AML, and so on, in addition to the existing classes ALL and AML. Furthermore, it is feasible to evaluate the efficacy of data augmentation by comparing the performance of traditional algorithms, such as rotation and shift, with GAN networks.”

Round 2
Reviewer 1 Report
Comments and Suggestions for Authors
Thank you for a successful revision. Some spelling errors continue in the study. It is possible to come across these errors even in the titles. For example, "4.1. optimization results". The title starts with a lowercase letter. These typos can be expanded upon. Please review your work more carefully.
Comments on the Quality of English Language.
Author Response
Thanks to the opinion of the respected referee, the relevant problems have been resolved.
Reviewer 3 Report
Comments and Suggestions for Authors
The authors solved all my criticisms
Author Response
Thank you for taking the time to review the article.